# Advanced Glycation End Products in Health and Disease

**DOI:** 10.3390/microorganisms10091848

**Published:** 2022-09-15

**Authors:** V. Prakash Reddy, Puspa Aryal, Emmanuel K. Darkwah

**Affiliations:** Department of Chemistry, Missouri University of Science and Technology, Rolla, MO 65409, USA

**Keywords:** advanced glycation end products, AGE inhibitors, AGE breakers, Alzheimer’s disease, diabetes, receptors for AGEs, traumatic brain injury, Maillard reaction, aminoguanidine, polyphenols

## Abstract

Advanced glycation end products (AGEs), formed through the nonenzymatic reaction of reducing sugars with the side-chain amino groups of lysine or arginine of proteins, followed by further glycoxidation reactions under oxidative stress conditions, are involved in the onset and exacerbation of a variety of diseases, including diabetes, atherosclerosis, and Alzheimer’s disease (AD) as well as in the secondary stages of traumatic brain injury (TBI). AGEs, in the form of intra- and interprotein crosslinks, deactivate various enzymes, exacerbating disease progression. The interactions of AGEs with the receptors for the AGEs (RAGE) also result in further downstream inflammatory cascade events. The overexpression of RAGE and the AGE-RAGE interactions are especially involved in cases of Alzheimer’s disease and other neurodegenerative diseases, including TBI and amyotrophic lateral sclerosis (ALS). Maillard reactions are also observed in the gut bacterial species. The protein aggregates found in the bacterial species resemble those of AD and Parkinson’s disease (PD), and AGE inhibitors increase the life span of the bacteria. Dietary AGEs alter the gut microbiota composition and elevate plasma glycosylation, thereby leading to systemic proinflammatory effects and endothelial dysfunction. There is emerging interest in developing AGE inhibitor and AGE breaker compounds to treat AGE-mediated pathologies, including diabetes and neurodegenerative diseases. Gut-microbiota-derived enzymes may also function as AGE-breaker biocatalysts. Thus, AGEs have a prominent role in the pathogenesis of various diseases, and the AGE inhibitor and AGE breaker approach may lead to novel therapeutic candidates.

## 1. Introduction

Nonenzymatic reactions of the terminal amino groups of amino acids or the side-chain amino groups of lysine and arginine in proteins with the carbonyl groups of reducing sugars, such as glucose, ribose, and trioses, result in the formation of Schiff bases that undergo further glycoxidation reactions when exposed to increased oxidative stress conditions to give the highly reactive 1,2-dicarbonyl compounds. These 1,2-dicarbonyl compounds react with the amino groups of the proteins, resulting in inter- and intraprotein crosslinks. The complex reactions involved in AGE formation are collectively also called Maillard reactions. The AGE-crosslinked proteins, including receptor proteins, and enzymes are thereby inactivated for normal physiological functions. These protein modifications and the small-molecule products resulting from their degradation are collectively called advanced glycation end products (AGEs; Figure 1) [1].

The formation of AGEs is promoted by elevated oxidative stress. The initially formed glycated proteins undergo extensive glycoxidation reactions involving reactive oxygen species (ROS), such as hydroxyl radical (HO·), and reactive nitrogen species (RNS), such as peroxynitrite anion (ONOO^−^). The AGEs, in turn, exert oxidative stress through their interaction with receptors for the advanced glycation end products (RAGE), which activates the inflammatory cytokine pathway. AGEs bind to various cell surface receptors, including receptors for AGEs (RAGE), oligosaccharyl transferase-48, 80K-H phosphoprotein, and galectin-3 in macrophages, monocytes, and microglia. The AGE modification of the nucleic acids, proteins, and enzymes impairs the biological functions of the receptors or enzymes, or activates them for the expression of proinflammatory cytokines, such as interleukin-6 (IL-6), which further leads to the exacerbation of the oxidative stress [2,3]. Therefore, inhibitors of AGEs (AGE inhibitors) or breakers of AGEs (AGE breakers), the small-molecule-based therapeutics or naturally occurring polyphenolic antioxidants, would attenuate AGEs, and the consequent AGE-related diseases would be attenuated or prevented (vide infra).

AGEs are also formed through nonenzymatic reactions of reducing sugars with the amino groups of the adenine, guanine, and aminolipids [4]. These endogenously formed AGEs, along with the ingested dietary AGEs, collectively contribute to the further oxidative stress and protein modifications and result in the activation of NF-kB, which modulates the expression of inflammatory genes, and thereby lead to the enhanced production of inflammatory cytokines [4,5]. The latter inflammatory pathways lead to the loss of cellular defense mechanisms and eventual neuronal cell death in AD. The excessive oxidative stress caused by these inflammatory pathways also contributes to the exacerbation of the other AGE-related pathologies, such as atherosclerosis and diabetes. Furthermore, the increased carbonyl stress and overproduction of reactive oxygen species (ROS) and reactive nitrogen species (RNS), under these oxidative stress conditions, causes the excessive production of AGEs, which may also lead to diabetes as well as pancreatic cancers [6].

Many of the AGEs are fluorescent compounds, and the extent of AGE formation is correlated with disease progression in AGE-related diseases, including diabetes; diabetic retinopathy and neuropathy [7,8,9,10]; cardiovascular diseases, including atherosclerosis [11,12,13,14,15]; and neurodegenerative diseases, such as Alzheimer’s disease (AD) [16,17,18,19], amyotrophic lateral sclerosis (ALS) [20,21,22,23], and Parkinson’s disease (PD) [24,25,26] as well as the secondary stages of traumatic brain injury (TBI) [27,28]. Most of the AGEs are not structurally characterized and are observed either through their characteristic fluorescence (λ_ex_ = 370 nm, λ_em_ = 440 nm) [29] or through immunocytochemistry using AGE-specific antibodies [30]. Some of the structurally characterized fluorescent AGEs include pentosidine, a fluorescent lysine–arginine crosslink; glucosepane, a nonfluorescent lysine–arginine crosslink; and nonfluorescent non-crosslinked AGEs, such as N^ε^-(carboxymethyl)lysine (CML) and argpyrimidine (Figure 1) [1,31,32].

It is widely recognized that, among other factors, AGEs are the predominant contributing factors in the onset and progression of diabetes and AD [1,4,33]. In some cases, type 2 diabetes leads to the onset of AD, and they both have common biomarkers, including AGEs, such as N^ε^-(carboxymethyl)lysine (CML) and pentosidine (a lysine–arginine crosslink; Figure 1) [16,17,18,19]. Hemoglobin A1_c_ is a CML-modified hemoglobin protein and is the routinely used clinical biomarker for diabetes. Through immunocytochemistry, it was shown that the CML is colocalized with lipid peroxidation products, such as malondialdehyde and 4-hydroxy-2-nonenal (HNE), in the neuronal cells in AD cases [16,17,18,19]. The latter findings suggested that CML is also derived through the lipid peroxidation pathway. However, it does not exclude the CML formation through the Maillard reaction sequence. The oxidative stress induced by AGEs also leads to the aggregation of the soluble amyloid-beta peptides (Aβ), and thus senile plaque formation contributes to a further accumulation of AGEs [34].

It is interesting that the Maillard reactions are also observed in bacterial species, including *E. coli*, and the protein aggregation in these bacterial cells is reminiscent of protein aggregates in Parkinson’s disease and AD [32]. Furthermore, dietary AGEs affect the gut microbiota composition, induce insulin resistance, and lower the abundance of the butyrate-forming gut bacteria [35]. Interestingly, AGE inhibitors extend the lifespan of the gut microbiota by attenuating AGE levels [36]. In model studies, *Escherichia coli* (*E. coli*), when exposed to glucose, decreased the life-span and health-span of its host nematode, *Caenorhabditis elegans* (*C. elegans*), and the physiological changes were accompanied by the dysregulation of glutathione-S-transferase and superoxide dismutase in the *C. elegans*. With chronic exposure to glucose, the bacterial species exhibited increased AGEs [37]. The exposure of bacterial species to the AGE inhibitor carnosine helped abrogate the negative effects of AGE-related effects in the nematode host [37]. Probiotic strains of *E. coli*, on the other hand, metabolize AGE compounds, such as N^ε^-carboxymethyllysine (CML), to form relatively nontoxic compounds (Figure 2) [38]. The MnmC bacterial enzyme (tRNA 5-aminomethyl-2-thiouridylate methyltransferase), which is involved in bacterial tRNA modifications, and its variants generated through site-directed mutagenesis were shown to reverse AGE modification in CML and N^ε^-carboxyethyllysine (CEL), effectively acting as AGE breaker enzymes [39]. MnmC is a bifunctional enzyme in which the C-terminal domain (C-MnmC) is a flavin adenine dinucleotide (FAD)-dependent methyl transferase and the N-terminal domain is a S-adenosylmethionine (SAM)-dependent methyl transferase. The C-terminal moiety of this enzyme, using FAD as a cofactor, brings about the oxidative deamination of α-amino acids to α-keto acids. This reaction goes through an α-iminimum carboxylic acid intermediate. Through a similar mechanism, the N-carboxymethyl and N-carboxyethyl moieties of CML and CEL are oxidatively hydrolyzed to form the lysine (Figure 2). 

Exogenous AGEs were shown to have a marked effect on the gut microbiota composition and plasma protein glycosylation. Thus, in mice, dietary AGEs lowered the populations of the *Lactobacillus*, *Prevotella*, *Anaerostipes*, and *Candidatus Arthromitus*, while the populations of *Parabacteroides*, *Ruminococcus*, and *Lawsonia* were elevated. The dietary AGE-mediated altered microbiota composition and elevated glycosylated plasma proteins are implicated in the systematic proinflammatory effects and endothelial dysfunction [40].

The Parkinson-associated DJ-1/PARK7-like protein glutamine amidotransferase-like class 1 domain-containing 3A (GATD3A) is a mitochondrial deglycase protein and has evolutionary origins from gammaproteobacteria. The mitochondrial GATD3A deglycase protein mediates the removal of early glycation intermediates, those derived from the Maillard reaction of glyoxal and methylglyoxal with the amino groups of nucleotides and amino acids, thereby maintaining the integrity of mitochondrial proteins. Mice lacking GATD3A are associated with elevated levels of AGEs and altered mitochondrial dynamics [41].

Thus, understanding the bacterial defense mechanisms against AGE formation would provide avenues for the development of novel therapeutics for the treatment of AD and other neurological disorders.

## 2. Receptors for Advanced Glycation End Products (RAGE)

Receptors for AGEs (RAGE) are a class of transmembrane multifunctional immunoglobulin superfamily of proteins. Figure 3 shows a schematic illustration of the structure of RAGE. RAGE exists in various isoforms, and generally it has extracellular C1, C2, and V domains, a transmembrane domain (TM), and a cytoplasmic domain. RAGE binds to a multitude of endogenous ligands (including AGEs and Aβ) via its extracellular as well as cytoplasmic domains [42].

The binding of ligands on the extracellular domains initiates a cascade of intracellular signaling events, leading to the production of reactive oxygen species (ROS) and inflammatory cytokines, resulting in cellular proliferation, cell apoptosis, and further upregulation of RAGE [42]. RAGE plays a major role in the onset of various pathological conditions, including cardiovascular diseases, neurodegeneration, cancer, and diabetes.

The extracellular domain of RAGE binds to AGEs, and this AGE-RAGE interaction leads to signal transduction through the activation of various kinases, such as mitogen-activated protein kinase (MAPKs), Janus kinase (JAK), phosphatidylinositol 3-kinase (PI3K), and the downstream activation of various inflammatory pathways, including the overexpression of nuclear factor kappa beta (NF-kB), a transcription factor for the expression of proinflammatory genes that regulate the formation of various inflammatory cytokines (IL-1, IL-6, and TNF-α) and thereby increase oxidative stress (overexpression of ROS and RNS) and cell apoptosis [42,43]. The AGE-RAGE interactions are implicated in the pathogenesis of various hepatic disorders, such as nonalcoholic steatohepatitis, liver cirrhosis, various cancers [44], AD [44], and cardiovascular diseases [45]. The levels of RAGE are relatively low in healthy individuals but are elevated under pathological conditions, including cancer, cardiovascular disease, and diabetes [46]. Targeting the extracellular and intracellular domains of RAGE is an emerging area in the development of RAGE-specific therapeutics [42,47].

CML and CEL (N^ε^-carboxyethyllysine) are abundant in the blood plasma and in tissues in diabetes and AD cases and are the major physiological ligands for RAGE. CML and CEL are formed through Maillard reactions of a lysine terminal amino group with glyoxal and methylglyoxal, respectively (Figure 4).

Solution protein NMR studies of the CEL-bound RAGE showed that the carboxyethyl and carboxymethyl moieties of CEL and CML bind to the positively charged cavity of the V domain of RAGE [48]. This binding of AGEs to RAGE is an indicator of, and in response to, cellular stressors. The activation of RAGE through its binding of AGEs leads to the expression of inflammatory cytokines in order to counteract the cellular stressors. However, the overexpression of various inflammatory cytokines, resulting from AGE-RAGE interactions, leads to a further increase in oxidative stress and to increased levels of AGEs, thereby propagating a vicious cycle that leads to cell apoptosis and disease severity. 

Maillard reactions of the guanidino moiety of arginine with methylglyoxal give three isomeric forms of AGEs, MG-H1 (N^δ^-(5-hydro-5-methyl-4-imidazolon-2-yl)ornithine), MG-H2 (5-(2-amino-5-hydro-5-methyl-4-imidazolon-1-yl)norvaline), and MG-H3 (5-(2-amino-4-hydro-4-methyl-5-imidazolon-1-yl)norvaline) (Figure 5). Of these isomers, MG-H1 is formed as the major isomer. These arginine-methylglyoxal-derived AGEs have a high affinity (K_d_ = 31 to 44 nM)) for the extracellular V-domain of RAGE. In comparison, the binding affinity of arginine to RAGE is too low because of the positive charge on its side-chain guanidino moiety, and therefore RAGE-arginine complexes were not observed [49]. Thus, RAGE selectively binds to the AGEs derived from the lysine and arginine side chains but not to lysine or arginine, as otherwise undesirable cellular signaling would be initiated upon such RAGE–amino acid binding. The RAGE-MG-H1 complex is stabilized by the hydrogen bonding interactions of the imidazolone moiety of MG-H1 with the surrounding positively charged amino moieties of Lys 32, Gln 80, and Lys 90 (Figure 6).

AGEs alone are inflammatory agents and cause increased oxidative stress, leading to damage to proteins and nucleic acids. Thus, the formation of AGEs and their interaction with RAGE initiates a vicious cycle of oxidative stress and the overproduction of AGEs. Soluble RAGE (sRAGE) comprises the extracellular fragments of the transmembrane protein RAGE that bind to AGEs, but because they are lacking the transmembrane protein, their binding to AGEs does not translate to the expression of inflammatory cytokines. The sRAGE are also secreted endogenously (called esRAGE). The sRAGE compete for the binding of the AGEs and thereby the AGE-RAGE interactions and the consequent cellular damage are circumvented. The latter sRAGE may provide avenues for designing therapeutics for various AGE-related diseases, including AD, the secondary pathological effects of TBI, and diabetes [50,51]. However, the clinical trials using various endogenous RAGE antagonists and the genetically engineered sRAGE have had little success to date [52].

RAGE antagonists are designed to target either the extracellular V domain of RAGE, which binds to the RAGE substrates (including AGEs and Aβ_1–42_), or the intracellular domain, which modulates RAGE signaling and downstream events; however, the majority of the potential RAGE antagonist therapeutics are based on targeting the extracellular V domain [47]. FPS-ZM1 (4-chloro-N-cyclohexyl-N-(phenylmethyl)benzamide; Figure 7), a RAGE antagonist, binds to the extracellular domain of the BACE and suppresses the AGE-induced expression of oxidative stress in rat primary microglial cells [53]. FPS-ZM1 suppressed the expression of NF-kB and downstream inflammatory mediators, such as TNF-α, interleukin-1 beta (IL-1β), cyclooxygenase 2 (COX-2), and inducible nitric oxide synthase (iNOS), and attenuated the AGE-induced formation of the NAPDPH oxidase (NOX) that activates the production of ROS. The latter RAGE antagonist also elevated antioxidant enzymes, such as heme oxygenase-1 (HO-1) [53]. It alleviated renal injury in hypertensive rats [54] and attenuated AGE-induced inflammation in the rat hippocampus [55].

Azeliragon (PF-04494700; Figure 7), a RAGE antagonist, competitively binds to the extracellular domain of RAGE, which has a relatively high affinity for RAGE ligands, including AGEs, HMGB1 (high-mobility group box 1 protein), and S100B (S100 calcium-binding protein B). Azeliragon also attenuates the levels of Aβ_1–42_. Although Azeliragon attenuated neuroinflammation by lowering the Aβ_1–42_ levels in a phase 2b clinical study [56], phase 3 clinical studies to evaluate the efficacy and safety of Azeliragon in patients with mild AD were discontinued because of the failure to achieve the primary endpoints. Other phase 3 trials of Azeliragon in patients with mild AD and impaired glucose tolerance are currently in progress [57]. A recent review summarized various RAGE antagonists that bind to the extracellular and intracellular domains of RAGE and their potential therapeutic efficiencies [42].

The FDA-approved thiazolidinedione class of drugs (Figure 7) for treating diabetes attenuate the AGE-RAGE interactions through an indirect mechanism: they inhibit the formation of RAGE and induce the expression of s-RAGE through the activation of the peroxisome proliferator-activated receptor gamma (PPAR-γ) [58,59].

Interestingly, in a retrospective cohort study in 362 patients with type 2 diabetes and 125 age- and gender-matched healthy control subjects for 15 years, there was a statistically significant correlation between the ratio of the circulating AGEs and soluble RAGE isoforms. In type 2 diabetes patients, circulating AGEs, total sRAGE, cRAGE (RAGE formed through cleavage from the extracellular surface of the RAGE), and AGEs/sRAGE and AGEs/esRAGE ratios were significantly increased compared to healthy controls. In healthy subjects, an inverse correlation of cRAGE and aging was observed, whereas in type 1 diabetes cases there was a positive correlation of these diagnostic markers. The increase in the AGEs/cRAGE ratio was accompanied by a high risk of all-cause mortality rates in type 2 diabetes. The increase in sRAGE was associated with the onset of cardiovascular diseases in type 2 diabetes cases [60]. Other studies also showed positive correlations of sRAGE levels with the onset of cardiovascular events in type 2 and type 1 diabetes [61,62]. Because the levels of RAGE, sRAGE, and cRAGE are directly related to the circulating levels of AGEs, a preventative strategy for disease pathogenesis would be to attenuate the AGE levels in diabetes cases through minimizing the intake of dietary AGEs (such as low consumption of processed foods) and through exercise-based lifestyle intervention [63,64,65].

The levels of plasma AGEs, sRAGE, NF-kB, and inflammatory markers were remarkably higher in type 2 diabetes patients with vascular complications and nephropathy [64]. sRAGE is also correlated with the severity of the chronic obstructive pulmonary disease (COPD) and it can be used as a biomarker for disease progression [66]. The high ratio of AGEs/sRAGE is also a risk factor for chronic kidney disease (CKD) [66].

The SARS-CoV-2-mediated cytokine storm is exacerbated in cases of diabetes, obesity, and high blood pressure in which AGEs are abundantly formed, indicating that the increased AGE-RAGE interactions and overexpressed RAGEs may be contributing factors for the severity of the disease in these cases [67]. In accordance with this hypothesis, a cross-sectional study of COVID-19 patients showed that there is a significant association between serum sRAGE and COVID-19 severity in severe COVID-19 cases [68]. In hamster models infected with the COVID-19 virus, it was shown that treatment with sRAGE attenuated the overactivation of inflammatory responses in SARS-CoV-2 [69], suggesting that targeting the AGE-RAGE axis may lead to pharmaceutical candidates for treating COVID-19 complications. In mouse models, the RAGE antagonist FPS-ZM1 (4-chloro-N-cyclohexyl-N-(phenylmethyl)benzamide) improved survival in infected mice, showing that impairing RAGE signaling would limit disease progression [70].

## 3. AGE Inhibitors

AGE inhibitors are naturally occurring polyphenolic compounds or small-molecule-based synthetic compounds, which act as AGE inhibitors through diverse modes of action, including acting as free-radical traps or antioxidants or through the sequestration of 1,2-dicarbonyl compounds (intermediate products of Maillard or lipid peroxidation reactions, leading to AGE formation). There is a wide-ranging interest in designing AGE inhibitors as potential therapeutics for a variety of AGE-related diseases, such as diabetes and diabetes-related neuropathy and retinopathy and neurodegenerative diseases, including Alzheimer’s disease (AD), Parkinson’s disease (PD), traumatic brain injury (TBI), and amyotrophic lateral sclerosis (ALS). Pyridoxamine (Vitamin B_6_) is an AGE inhibitor as well as a lipid peroxidation inhibitor and attenuates the levels of AGEs [71,72,73,74]. Pyridoxamine ameliorated the complications from diabetic retinopathy in experimental rat models [75] and therefore is potentially useful in the treatment of human diabetes or diabetic retinopathy. However, safety concerns during the clinical trials impeded the use of pyridoxamine as a therapeutic candidate in diabetic nephropathy [6]. Carnosine, β-(alanyl)histidine, is an antioxidant compound, transition metal ion chelator, and scavenger of reactive 1,2-dicarbonyl compounds and therefore has antiglycating effectiveness (Figure 8) [76,77,78]. However, the relatively fast hydrolytic cleavage of carnosine by the intracellular carnosinase enzyme limits its use as a therapeutic AGE inhibitor [76,79]. The glyoxalase enzymes, glyoxalase I and glyoxalase II, use reduced glutathione as a cofactor in the detoxification of AGEs. The levels of the glyoxalase I correlate with decreases in AGE content, and the downregulation of glyoxalase I is associated with increased levels of AGEs [4,80]. In this glyoxalase-catalyzed detoxification of methylglyoxal, the reaction of methylglyoxal with glutathione initially gives a thiohemiacetal, which is then metabolized to the nontoxic D-lactate [81]. 

2-Isopropylidenehydrazono-4-oxo-thiazolidin-5-ylacetanilide (OPB-9195; Figure 8) was developed as a potential drug candidate for treating diabetes complications. The latter compound lowers the levels of AGEs as well as advanced lipoxidation end products (ALE), such as 4-hydroxy-2-nonenal (HNE), a toxic α,β−unsaturated aldehyde that undergoes Michael additions to amino or thiol moieties of various proteins and deactivates their enzymatic functions [82].

Aminoguanidine (pimagedine), as an AGE inhibitor, has received more attention than any other AGE inhibitors. Polyphenolic compounds have multiple mechanisms of action in attenuating the levels of AGEs and, as naturally occurring compounds, have no adverse effects, unlike synthetic AGE inhibitors, including aminoguanidine. The following sections outline the role of aminoguanidine and polyphenolic compounds as AGE inhibitors and their effectiveness in treating AGE-related diseases, including diabetes and AD.

### 3.1. Aminoguanidine

Aminoguanidine was among the earliest AGE inhibitors that went into clinical trials to treat diabetes and diabetes-related complications, such as diabetic retinopathy. However, these clinical trials were withdrawn due to the adverse effects of aminoguanidine in the phase II/III clinical trials [83,84,85]. Thus, in randomized, double-blinded, placebo-controlled clinical trials with 690 patients with diabetes mellitus type 1 (type 1 diabetes), diabetic nephropathy, and diabetic neuropathy, aminoguanidine provided clinical proof that inhibiting AGEs may ameliorate the complications of diabetes. However, patients receiving high-dose aminoguanidine exhibited glomerulonephritis, and furthermore, these studies did not demonstrate statistically significant beneficial effects on diabetic nephropathy [86,87]. These clinical studies and others on AGE inhibitors and AGE breaker compounds, in addition to their toxicity concerns, show marginal effects on the mitigation of diabetes or diabetes-related complications [73]. However, the potential beneficial effects of aminoguanidine and other AGE inhibitors and AGE breakers in mitigating various pathological conditions are substantially supported by animal model studies [73,88,89,90,91,92,93,94,95,96,97,98,99,100]. AGE inhibitors, including aminoguanidine and pyridoxamine, trap not only the reactive 1,2-dicarbonyl compound intermediates of the Maillard reaction but also the lipid precursors of advanced lipid peroxidation end products (ALE), thereby showing protective effects in the development of atherosclerosis, early renal disease, and dyslipidemia in animal models [101,102].

Due to the high reactivity toward nucleophilic addition reactions, aminoguanidine traps 1,2-dicarbonyl compounds, such as methylglyoxal, glyoxal, glucosone, and dehydroascorbate, at relatively rapid rates. The latter 1,2-dicarbonyl compounds are highly electrophilic and are relatively more reactive than reducing sugars, such as D-glucose, under physiological conditions. Thus, the trapping of intermediate 1,2-dicarbonyl compounds precludes their further transformation to AGEs. The excess accumulation of the 1,2-dicarbonyl compounds is also called carbonyl stress, which is attenuated by aminoguanidine [103] (Figure 9) [104,105]. The reactive 1,2-dicarbonyl compounds are also generated through lipid peroxidation pathways (and through the cellular metabolism of lipids) in addition to their formation as early Maillard reaction products. The carbonyl stress arising from unhealthy processed food, thus, is also a major contributing risk factor for the onset of cardiometabolic and cancer pathologies [6,106]. Both type 1 and type 2 diabetes are major risk factors for pancreatic cancer, and importantly all these pathologies have excessive accumulation of AGEs and elevated carbonyl stress. Furthermore, the glyoxalate pathway, which degrades excessive methylglyoxal in diabetes cases, is impaired, contributing to carbonyl stress [107]. Glycemic control as well as the attenuation of oxidative stress are therefore key parameters for designing naturally occurring or synthetic therapeutics, such as aminoguanidine-based AGE inhibitors.

The AGE inhibitor compound aminoguanidine prevents decreased myocardial compliance in streptozotocin-induced diabetic rats [108]. The myocardial collagen AGE fluorescence intensity is decreased upon the administration of the aminoguanidine in these cases, which shows that AGEs contribute to myocardial stiffness. Aminoguanidine treatment increases vascular elasticity and decreases vascular permeability in diabetic rat models and may exert a mitigating effect on diabetic cardiomyopathy and diabetic heart failure [103,109]. Aminoguanidine prevents arterial stiffening in diabetic rat models, which is attributed to the AGE-inhibitory effect of aminoguanidine [110,111]. Thus, AGE accumulation has deleterious effects on vascular collagen crosslinking, which, in turn, increases the arterial wall stiffness and the permeability to fluids. Therefore, AGE inhibitors have a positive outcome in these cases, as demonstrated in animal models. 

Aminoguanidine is also an inhibitor of inducible nitric oxide synthase (iNOS), thereby attenuating the levels of nitric oxide (NO) [112]. Thus, the oxidative stress markers of reactive nitrogen species (RNS), including NO and peroxynitrite (ONOO^−^), are attenuated by aminoguanidine, and the levels of the AGEs are thereby attenuated [112].

### 3.2. Pyridoxamine

Pyridoxamine (Figure 8), due to the presence of the phenolic hydroxy group, can act as a free-radical scavenging agent, thus sequestering the ROS and RNS and thereby suppressing oxidative stress and AGE formation. Pyridoxamine inhibits the generation of hydroxyl radical from the albumin–Amadori system and thereby protects the albumin–Amadori-induced tryptophan modification and exerts its protective effects on diabetes complications, including diabetic nephropathy [113,114]. Pyridoxamine was also shown to affect wound healing in nonhealing diabetic wounds through the scavenging of methylglyoxal, thereby inhibiting the methylglyoxal-mediated formation of the protein adducts that cause macrophage dysfunction [115].

In phase 2 clinical studies with type 1 and type 2 diabetes and early-stage diabetic nephropathy, pyridoxamine attenuated the levels of CML and carboxyethyllysine (CEL) AGEs, although these studies did not give confirmatory evidence of its beneficial effect in treating diabetic nephropathy [87,116]. The combined effect of the dietary AGEs and endogenously formed AGEs contributes to chronic kidney diseases, including diabetic nephropathy. Kidneys are vital in the clearance of AGEs, and the excessive accumulation of AGEs, especially in diabetic patients, results in exacerbated pathology and a progression to end-stage diabetic nephropathy [8]. Thus, AGE clearance by AGE inhibitors or AGE breakers would be expected to lead to potential therapeutics. 

Pyridoxamine was shown to be a better antiglycating agent compared to metformin in the early, intermediate, and late stages of glycation [117]. In clinical studies, pyridoxamine, when co-administered along with methylcobalamin and benfotiamine, helped decrease pain and inflammation in osteoarthritis and rheumatoid arthritis patients [118,119]. Pyridoxamine as well as aminoguanidine attenuated AGE (generated through the Maillard reaction of glyceraldehyde)-mediated tau-protein phosphorylation and β-tubulin aggregation and suppressed glyceraldehyde-AGE-induced dysfunctional neurite outgrowth, showing the potential therapeutic effects of the AGE inhibitor compounds, such as pyridoxamine and aminoguanidine [120]. Through the suppression of the methylglyoxal-mediated AGEs, pyridoxamine was shown to revert the methylglyoxal-induced loss of cell survival pathways in ischemia [121]. Pyridoxamine may also have potential therapeutic effects in treating irritable bowel syndrome (IBM), as it attenuates the fermentable-carbohydrate-mediated AGEs and thereby prevents IBM. The latter gastrointestinal disorder is associated with increased fermentable carbohydrate intake, and thus AGEs are implicated in this negative impact of fermentable carbohydrates [122]. Pyridoxamine suppresses the methylglyoxal-mediated formation of AGEs, including argpyrimidine, and prevented the apoptosis of methylglyoxal-treated human lens epithelial cells [123]. Pyridoxamine, by inhibiting the formation of AGEs, attenuates diabetic complications [123].

Sphingolipid metabolism is impaired in the case of insulin-resistant mice, and this impaired sphingolipid metabolism is attributed to AGE accumulation in liver and the associated AGE/RAGE signaling pathways [124]. Pyridoxamine inhibits the formation of AGEs and thereby prevents sphingolipid alterations and attenuates insulin resistance [124]. AGEs are also involved in the mitigation of liver fibrosis, a common chronic hepatic disease. Pyridoxamine, in animal models, reduces liver fibrosis, presumably through its AGE-inhibitory effect and the attenuation of oxidative stress [125].

### 3.3. Polyphenols as AGE Inhibitors

Polyphenolic antioxidants inhibit the formation of AGEs through a free-radical scavenging effect and transition metal ion (e.g., Cu^+^ and Fe^2+)^-chelating effects. They also activate insulin signaling pathways, promoting glucose metabolism [126,127,128]. The phenolic oxy radicals are stabilized by resonance and are more stable than ROS. Therefore, polyphenolic compounds effectively quench ROS. In this process, the hydroxyl radical is transformed to the hydroxide anion, and the phenolic compounds are transformed to phenoxy radicals. The polyphenols also sequester RNS, such as peroxynitrite anion (ONOO^−^), through electrophilic or free-radical aromatic nitration reactions [44].

RNS and ROS are formed through Fenton reactions involving the reaction of transition metal ions, such as Cu^+^ and Fe^2+^, with the molecular oxygen, initially forming the superoxide radical anion (O_2_^−^), which undergoes further metal-ion-catalyzed redox reactions to give the highly reactive hydroxy radicals (HO·). The reaction of superoxide radical anion with nitric oxide NO (a metabolic product of nitric oxide synthase) forms the highly oxidizing RNS peroxynitrite anion (ONOO^−^) [1,127]. Thus, through metal ion chelation by polyphenolic compounds, the formation of ROS and RNS is suppressed and thereby the levels of AGEs are attenuated. 

Metal ion chelation by ethylene diamine-N,N,N,N-tetraacetic acid (EDTA) attenuates the formation of ROS and RNS and thereby attenuates the glycoxidation reactions and the AGEs [129]. In clinical trials, EDTA showed some positive outcomes in the treatment of atherosclerosis and diabetes, although the chelation therapy involves an invasive procedure [130,131,132]. It was hypothesized that the AGE-inhibitory effects of some of the commonly used drugs to treat diabetes, such as ACE inhibitors, angiotensin receptor blockers, and aldolase reductase inhibitors, may be due to their chelating effect on metal ions [44,129]. These drugs do not have functional groups that would trap the reactive 1,2-dicarbonyl Maillard intermediate products or that would trap free radical species.

Randomized clinical trials showed that resveratrol, a polyphenolic compound, ameliorates, to some extent, the progressive cognitive decline in AD patients [34]. Many other studies demonstrated that resveratrol has neuroprotective, anti-inflammatory, and antioxidant properties and that it can activate silent information regulator-1 (SIRT1) and thereby protect neuronal cells from the toxic effects of excessive oxidative stress [133]. Furthermore, resveratrol inhibits the expression of AGEs and thereby decreases the insoluble Aβ_1–42_ levels in AD brains and protects the blood–brain barrier (BBB) [134,135,136]. However, resveratrol has poor bioavailability and relatively unfavorable pharmacokinetics [120], thus impeding its AGE-inhibitor therapeutic effectiveness in treating AGE-related diseases, including diabetic nephropathy.

The levels of AGEs are attenuated in the presence of polyphenolic compounds, including resveratrol, curcumin, and flavonoids, such as epigallocatechin gallate and catechin, and thus polyphenolic compounds have neuroprotective effects [2]. However, effective therapeutics based on polyphenolic compounds for treating neurodegenerative diseases are lacking to date. Antioxidants, such as ascorbic acid, glutathione, and vitamin E (α-tocopherol) also exert antiglycating and neuroprotective effects. Pomegranate-derived polyphenolic compounds, such as ellagic acid and gallic acid, and their intestinal bacterial metabolites, urolithin A and urolithin B, show an antiglycating effect and were shown to be more potent AGE inhibitors compared to aminoguanidine (Figure 10) [137]. These polyphenolic compounds, through the attenuation of ROS and RNS, inhibit the formation of AGEs. 

## 4. AGE Breakers

AGE breaker compounds reverse the protein crosslinks in AGEs, thereby releasing the proteins in their natural form and reversing the toxic effects associated with AGEs. Alteon, Inc. has developed phenacyl thiazolium-based compounds, such as Alagebrium (ALT-711) and phenacylthiazolium bromide (PTB), as potential therapeutic candidates (Figure 11) [138,139,140,141,142,143]. Alagebrium, in phase III clinical tries, ameliorated the stiffening of the arterial vessels and reduced systolic blood pressure in patients with diastolic heart failure [144,145]. This drug candidate also has favorable outcomes in the treatment of diabetes and hypertension and has beneficial effects on other AGE-related diseases; however, the clinical trials were later abandoned due to other issues unrelated to the safety profile. 

The AGE breaker compound TRC-4149 (Figure 12) was studied in vitro and in vivo in streptozotocin-induced diabetic spontaneously hypertensive rats (SHR) [146]. The latter AGE breaker compound was shown to break the AGE crosslinks and to reduce the AGE burden in a dose-dependent manner. In the SHR animal models, it improved cardiac function compared to the controls and reduced the AGE load, preserving the endothelial and cardiac functions. TRC-4149 exhibits favorable outcomes in reversing diabetic complications in in vitro and in vivo studies, and in the phase I clinical trials, showed favorable safety profiles when administered orally either as a single dose or multiple doses [147,148]. However, despite these efforts in developing novel therapeutic candidates, there has been no successful AGE breaker candidate that can be used as a pharmaceutical to date.

Alagebrium breaks methylglyoxal-mediated AGE protein crosslinks. However, its AGE-breaking efficiency toward the AGEs formed from the other 1,2-dicarbonyl compounds (such as glucosone) is not well-demonstrated [145,149]. The mechanism of the AGE-breaking reaction is not well-established [150]. Using model systems of α-ketoaldehydes, it was shown that ALT-711 forms cyclic diol products and scavenges methylglyoxal under physiological conditions [149]. This proposed mechanism is in accordance with that of Vasan and coworkers’ earlier proposed mechanism for the PTB-mediated degradation of the protein-crosslinked 1,2-dicarbonyl compounds [138,150].

The use of the phenacylthiazolium-based compounds, such as ALT-711, may be more specific to the cleavage of 1,2-dicarbonyl-containing protein crosslinks under physiological conditions. However, the AGE breaker compound TRC-4149 lacks such N-heterocyclic carbene (NHC) functionality to form the cyclic diol intermediates with α-ketoaldehydes. As an alternative mechanism that accounts for the AGE-breaking effect of thiazolium as well as pyridinium-based AGE breakers, shown in Figure 12, we propose that the AGE-breaking reaction may involve the nucleophilic addition of the enolate anion of the AGE breaker compounds (including ALT-711 and TRC-4149) to the imine moiety of the AGE protein crosslinks, followed by the cleavage of the C-C bond connecting the imino moieties. Thus, generated protein-bound imines are further hydrolyzed, in situ, to give AGE-free proteins (Figure 12). Further experiments using various related enolate anions as model AGE breaker compounds may provide additional evidence for this proposed mechanism.

The thiazolium moiety in PTB and Alagebrium and the pyridinium moiety in TRC-4149 may also selectively bind to the AGE crosslinks, similar to the selective amyloid binding to thioflavin T [151,152], thereby facilitating the cleavage of the crosslink imine moieties. Further research is needed for the elucidation of the mechanism of the AGE-breaker-mediated degradation of AGEs. However, the AGE breaker compounds, such as alagebrium and PTB, seem to act specifically on methylglyoxal-mediated AGEs [150]. The delineation of the AGE breaker and AGE inhibitor mechanisms would enable the design of AGE breakers that are applicable to the degradation of a broader class of AGE crosslinks.

AGEs induce collagen crosslinking to a greater extent than in the control cases in the absence of AGEs. Thus, increased collagen crosslinks translate to the stiffening of arterial walls and the accompanying atherosclerosis. The AGE crosslink breaker ALT-711 was shown to reverse protein crosslinking and the diabetes-induced increase in arterial stiffness in the streptozotocin-induced diabetes rat models [153]. An ester analog of PTB, 3-benzyloxycarbonylmethyl-4-methyl-3-thiazolium bromide (C-36; Figure 10), improves the cardiovascular system of diabetic rats and attenuates the mRNA levels of diabetes-induced genes, including RAGE [154]. 

The AGE breaker compound ALT-711 reverses arterial and ventricular wall stiffness in healthy older rhesus monkeys [155]. It was shown that in diabetic hypertensive rats, the coadministration of ALT-711 with the Ca^2+^ channel blocker nifedipine improves the antihypertensive efficacy [156]. These studies have an impact in AGE-breaker-based drug discovery for the heart failure associated with arterial and ventricular wall stiffening in diabetes and hypertension cases. A randomized human clinical trial of ALT-711, during a period of one year, however, showed no significant effect on arterial stiffness [157]. The initial clinical trials of alagebrium from 2002 to 2010 (Synvista Therapeutics) provided data that showed a decrease in arterial pulse pressure and endothelial function in hypertension patients. However, a separate clinical trial showed no improvement in diastolic or systolic function or AGE accumulation [158].

In a randomized placebo-controlled clinical study, alagebrium did not improve exercise tolerance in patients with heart failure and systolic function and thus had no beneficial effect in systolic heart failure, although it was well-tolerated by patients [73,159]. However, other clinical trials using ALT-711 showed a decrease in left ventricular mass and improvements in left ventricular diastolic filling in diabetic heart failure patients [144].

## 5. Conclusions and Outlook

AGEs play a major role in the pathogenesis of diabetic complications, such as diabetic neuropathy and nephropathy, and in the onset of neurological disorders, including AD, PD, and TBI-mediated AD-related dementia (ADRD). Figure 13 outlines the formation of AGEs; the AGE inhibitor-mediated attenuation of oxidative stress and carbonyl stress and the attenuation of AGE levels; and the AGE-breaker-mediated degradation of AGEs. Interactions of AGEs with RAGE initiate cellular signaling, thereby activating nuclear factors, such as NF-kB, which are involved in the expression of inflammatory cytokines (such as interleukin-6) and the induction of further oxidative stress. 

AGE protein crosslinks and small-molecule AGEs, including dietary AGEs, through AGE-RAGE interactions, are involved in the onset and exacerbation of various AGE-related gastrointestinal diseases and neurological disorders, such as AD and TBI-associated AD-related dementia (ADRD). The dietary and endogenously formed AGEs affect the gut microbiota composition and insulin resistance. In addition to the binding to the RAGE, AGEs also bind to the cell surface receptors of macrophages and other AGE-specific receptors and are thereby degraded to nontoxic compounds. 

Although many pathologies outlined in Figure 12 have multifactorial origin, AGEs and AGE-related oxidative stress are major contributing factors in the onset and exacerbation of various diseases, collectively called AGE-related diseases. Therefore, there is an enormous interest in identifying safer and effective AGE inhibitors to treat these AGE-related diseases. Despite substantial efforts in developing novel drugs based on the AGE inhibitor or AGE breaker concept, there have been no FDA-approved therapeutics that can be used in the treatment of the AGE-induced pathologies, such as diabetes and AD, even though AGE inhibitors, such as aminoguanidine and pyridoxamine, exhibited therapeutic potential in animal models. There is also renewed interest in developing safer and relatively effective versions of the AGE breaker compounds to lower AGE levels. Polyphenolic compounds are relatively safe alternatives to AGE inhibitors, although there have been no wide-ranging clinical trials of using such compounds to demonstrate their effectiveness in the treatment of diabetes or neurological disorders. The mechanisms of formation of AGEs may be similar in intestinal bacteria and humans, as the AGE-mediated protein aggregates in intestinal bacterial species have features in common with those in neurological disorders. Thus, intestinal bacterial defense mechanisms toward AGE-mediated pathogenesis may shed new light on the future AGE inhibitor and AGE breaker therapeutic candidates.

## Figures and Tables

**Figure 1 microorganisms-10-01848-f001:**
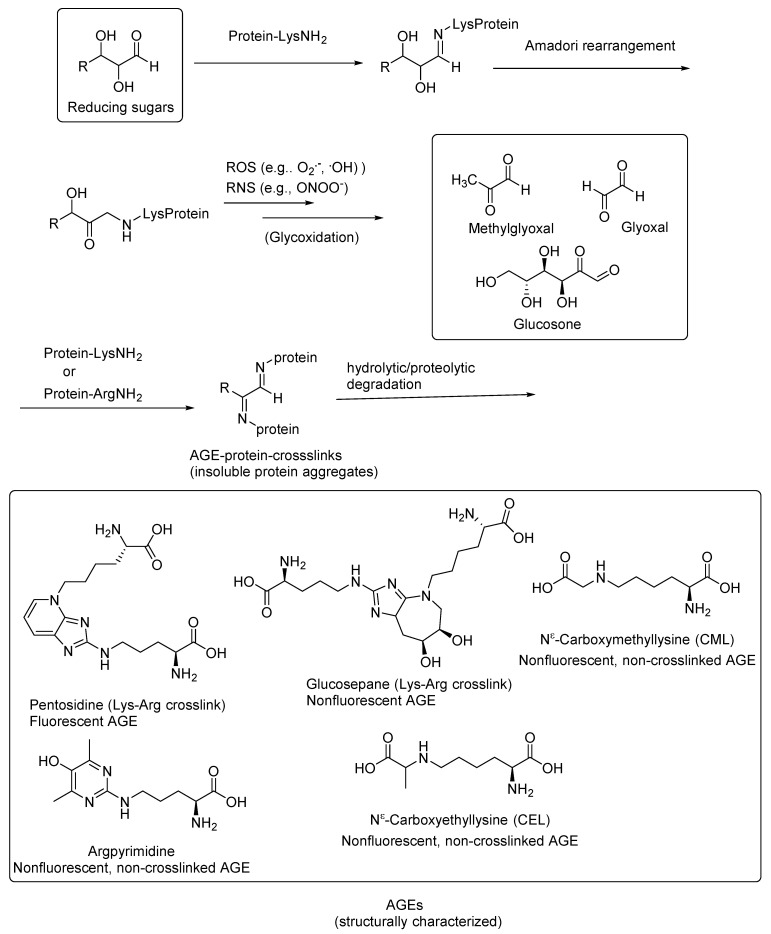
Formation of AGEs through nonenzymatic reactions of reducing sugars with protein amino groups.

**Figure 2 microorganisms-10-01848-f002:**
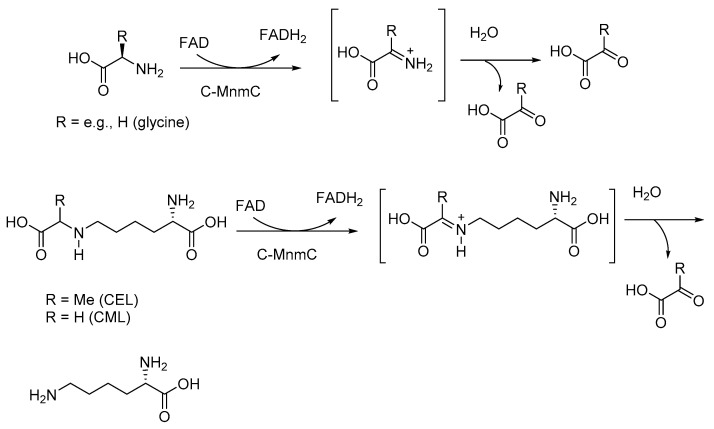
MnmC-catalyzed oxidative deamination of α-amino acids and reversal of AGE modifications of CML and CEL.

**Figure 3 microorganisms-10-01848-f003:**
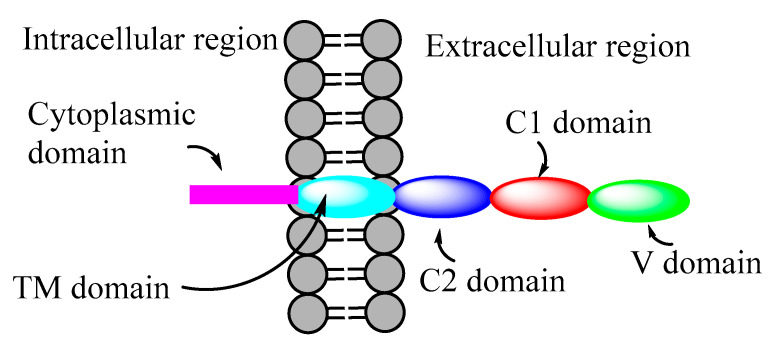
Schematic illustration of the structure of RAGE, showing extracellular, transmembrane, and cytoplasmic domains.

**Figure 4 microorganisms-10-01848-f004:**
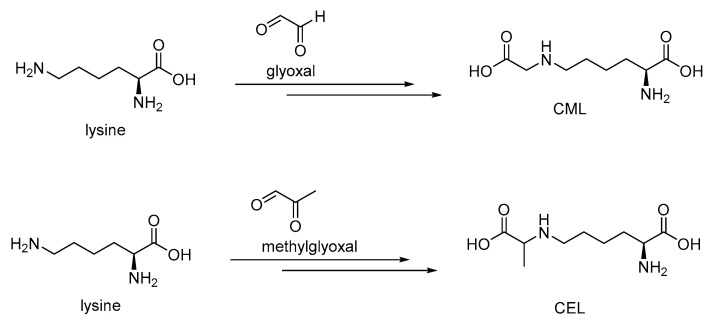
Formation CML and CEL through Maillard reactions.

**Figure 5 microorganisms-10-01848-f005:**
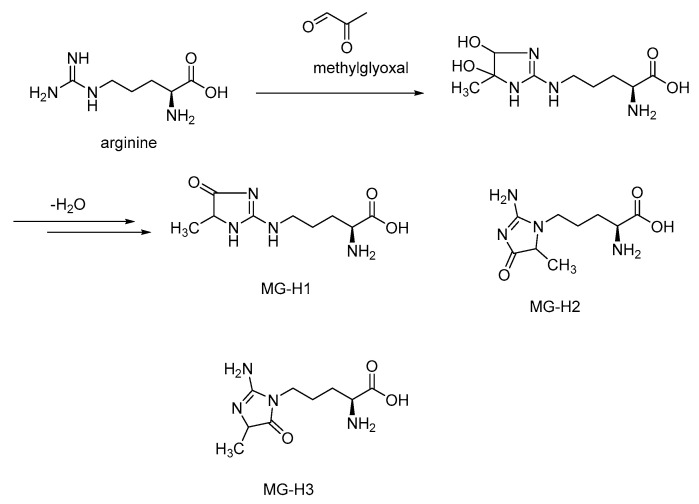
Maillard-reaction-derived AGE products of arginine and methylglyoxal.

**Figure 6 microorganisms-10-01848-f006:**
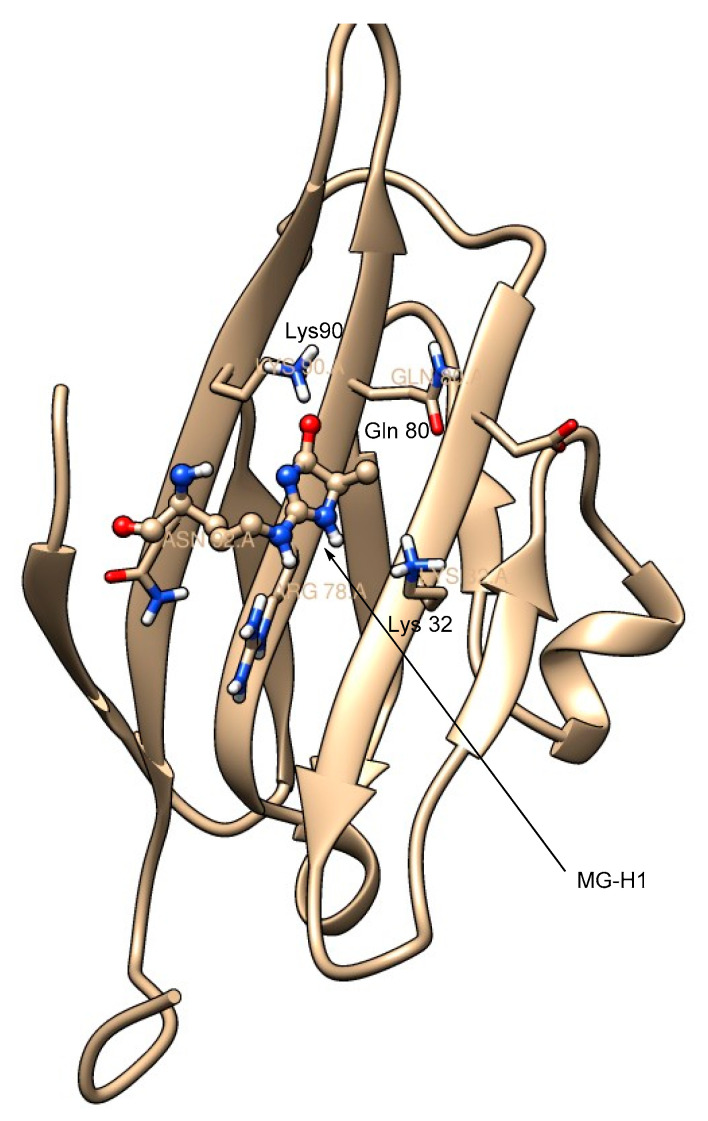
Solution NMR structure of the RAGE-MG-H1 complex (shown here is the extracellular V-domain of the RAGE bound to MG-H1); the imidazolone moiety of MG-H1 is surrounded by the positively charged amino acid residues, including Lys 90, Gln 80, and Lys 32; the structure was created using UCSF Chimera software; PDB ID: 2MOV (MG-H1 is shown as a ball-and-stick model; red = oxygen, blue = nitrogen).

**Figure 7 microorganisms-10-01848-f007:**
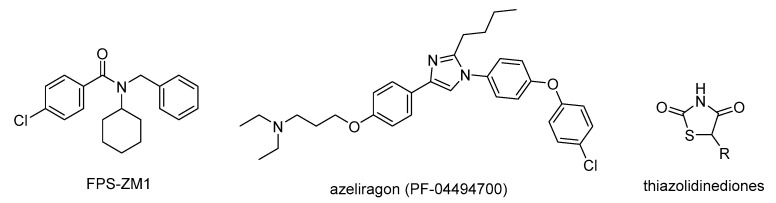
Structures of RAGE antagonists FPS-ZM1 and Azeliragon and the antidiabetic drug thiazolidinedione, which inhibits the formation of RAGEs.

**Figure 8 microorganisms-10-01848-f008:**
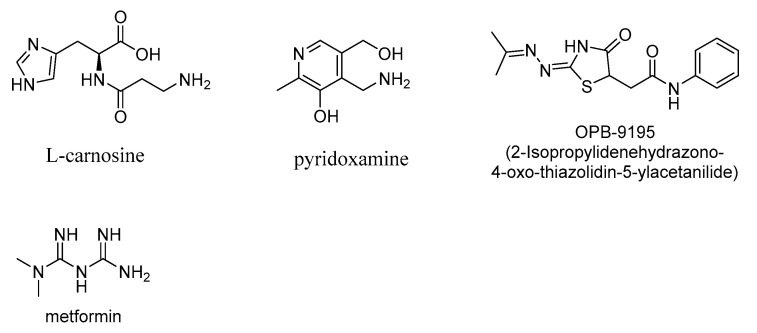
Structures of AGE-inhibitor compounds L-carnosine, pyridoxamine, and metformin.

**Figure 9 microorganisms-10-01848-f009:**
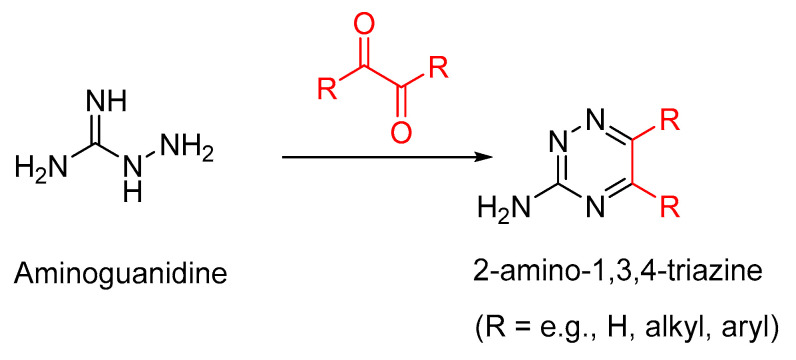
Sequestration of the reactive 1,2-dicarbonyl compounds, the early Maillard reaction intermediates, by aminoguanidine.

**Figure 10 microorganisms-10-01848-f010:**
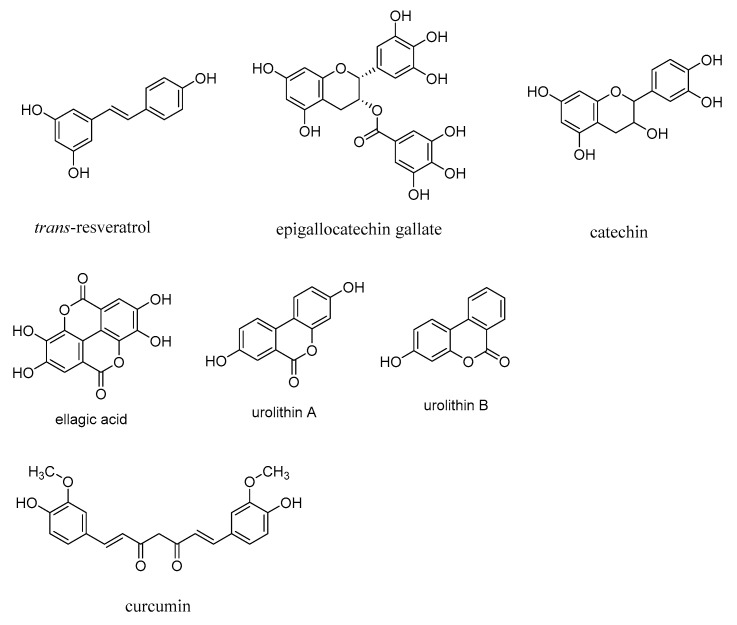
Structures of polyphenolic antioxidants, *trans*-resveratrol, curcumin, catechin, epigallocatechin gallate, ellagic acid, urolithin A, and urolithin B.

**Figure 11 microorganisms-10-01848-f011:**
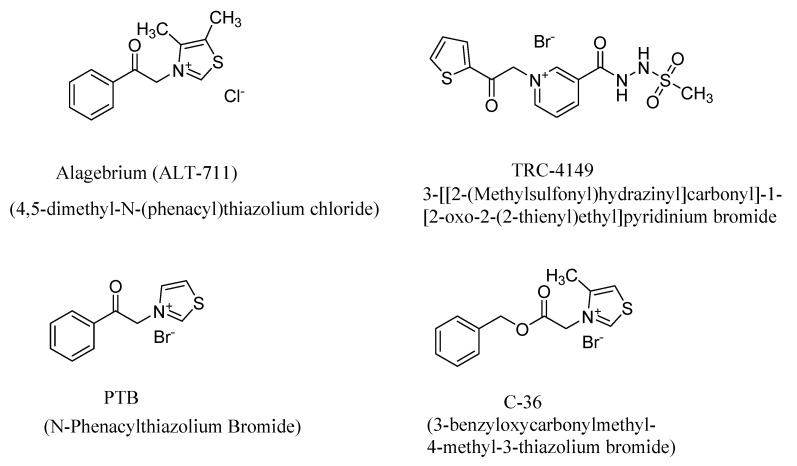
Structures of the AGE breaker compounds.

**Figure 12 microorganisms-10-01848-f012:**
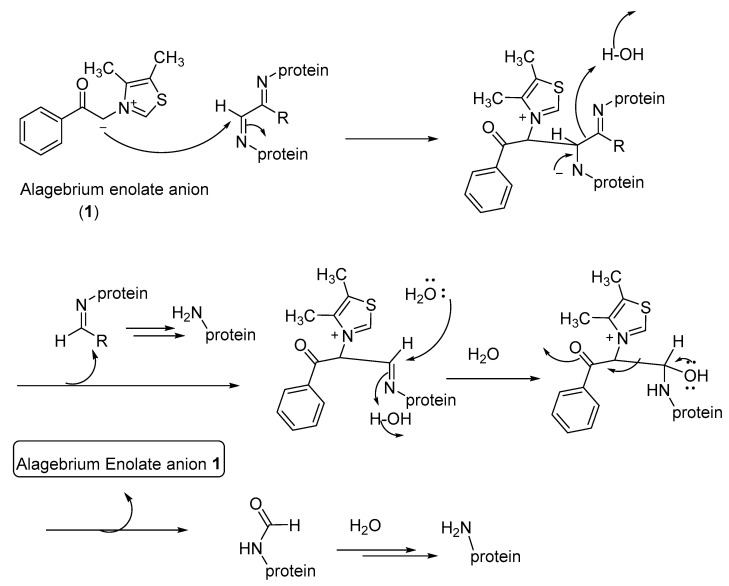
Proposed mechanism for the AGE-breaker-mediated hydrolysis of protein crosslinks.

**Figure 13 microorganisms-10-01848-f013:**
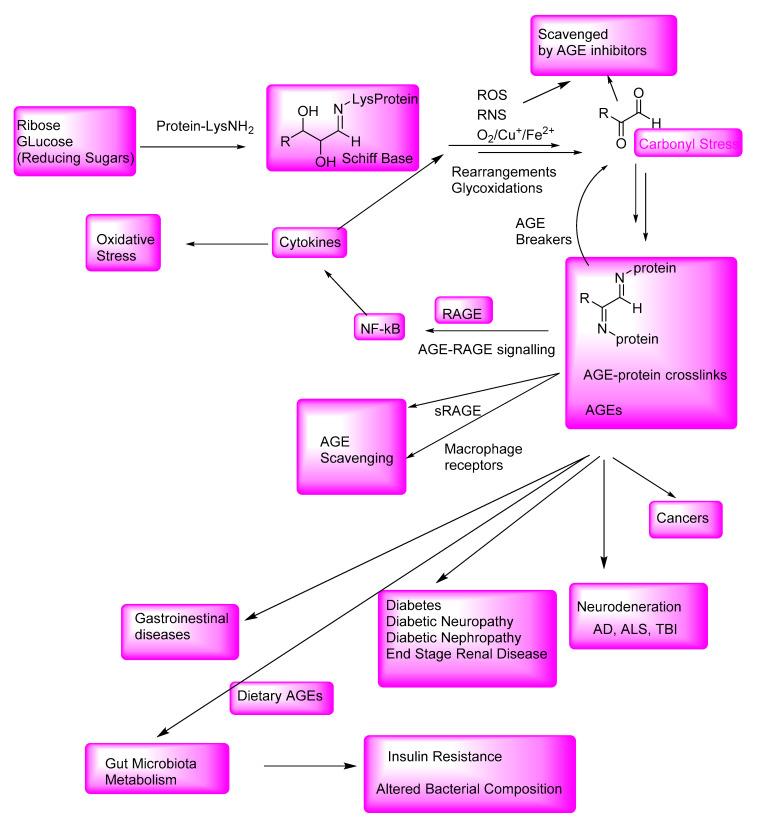
Schematic outline summarizing the formation of AGEs, AGE inhibitors, and AGE breakers, AGE-RAGE interactions, and the onset and exacerbation of various AGE-related diseases.

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
