# Peer review of "Advanced Glycation End Products in Health and Disease"

_microorganisms, 2022, doi:10.3390/microorganisms10091848_

Round 1

Reviewer 1 Report

The paper titled “Advanced Glycation End Products in Health and Disease” is a well written, informative literature review report regarding Advanced Glycation End Products (AGEs), their receptors (RAGEs) and their involvement in disease, as well as some therapeutic agents, their mechanism of action and their potential contribution to patients with AGEs-related diseases.

The paper describes the formation, structure and function of AGEs and RAGEs and their role in the onset and progression of some diseases, such as diabetes and Alzheimer’s disease. The authors also discuss the distinction between endogenous and dietary AGEs as they explain exposure to these agents. Furthermore, the authors discuss the advantages of soluble RAGE (sRAGE), which is a soluble AGE receptor that competes for AGE binding and prevents the negative consequences of AGE-GARE binding, as sRAGE is not bound to the cell membrane and, therefore, is not part of the detrimental pathways which eventually lead to the aforementioned diseases.

The authors go on to discuss several agents which present therapeutic potential:

·       RAGE antagonists – which bind to RAGEs and prevent the downstream AGE-RAGE signaling.

·       AGE inhibitors – which prevent AGEs from exerting their detrimental effects in several ways.

·       AGE breakers – which affect the formation of AGEs and have similar effects as the AGE inhibitors.

The paper is comprehensive and the images efficiently illustrate the points made by the authors. One editing comment:

·       Line 100: the title for #2 is “Receptors for Advanced Glycation Inhibitors (RAGE)”. The authors’ intention may have been “Receptors for Advanced Glycation End Products (RAGE)”

Author Response

We appreciate the reviewer’s encouraging comments.  We also appreciate the reviewer for his careful reading of the manuscript and for pointing out the error in line 100; We have now corrected the typographical error: “Receptors for Advanced Glycation End Products (RAGE)”.

Reviewer 2 Report

Advanced Glycation End Products in Health and Disease

The article is well written and easy to read but I have several comments.

1.      Abstract. The authors should explain the abbreviation "PD" (Parkinson Disease?).

2.      The subchapter 2: „Receptors for Advanced Glycation Inhibitors (RAGE)“ is somewhat misleading and need rephrasing (e.g. Receptors for Advanced Glycation End Products and their antagonist?). For the benefit of the non-specialist reader, please, explain briefly the structure of RAGE.

3.      Figure 10, please, explain meaning “I”

4.      There are many places in the text where spaces are repeated. Please remove it.

5.      There are many sentences in the text that are too long and difficult to understand. Please edit them

For example Lane 170-173

“RAGE antagonists are designed to target either the extracellular V domain of the

RAGE, which bind to the RAGE substrates, including AGEs and A1-42, or the intracellular domain, which modulates the RAGE signaling and downstream events, although a majority of the potential RAGE antagonist therapeutics are based on targeting the extracellular domain.”

6.      Lane 488 “collage” – should be “collagen”?

Author Response

  1. The authors should explain the abbreviation "PD" (Parkinson Disease?).

We have now expanded the abbreviation as follows: “Parkinson’s disease (PD)”.

  1. The subchapter 2: „Receptors for Advanced Glycation Inhibitors (RAGE)“ is somewhat misleading and need rephrasing (e.g. Receptors for Advanced Glycation End Products and their antagonist?). For the benefit of the non-specialist reader, please, explain briefly the structure of RAGE.

We appreciate the reviewer for his careful reading of the manuscript and pointing out he error; we have corrected the typographical error as follows: “Receptors for Advanced Glycation End Products (RAGE).” We have also briefly summarized the structural details of RAGE.

  1. Figure 10, please, explain meaning “I”

We appreciate the reviewer for pointing it out; we have now clarified it by replacing “I” to “Alagebrium enolate anion (1)”.

  1. There are many places in the text where spaces are repeated. Please remove it.

We have corrected it.

  1. There are many sentences in the text that are too long and difficult to understand. Please edit them

For example Lane 170-173

“RAGE antagonists are designed to target either the extracellular V domain of the

RAGE, which bind to the RAGE substrates, including AGEs and Ab1-42, or the intracellular domain, which modulates the RAGE signaling and downstream events, although a majority of the potential RAGE antagonist therapeutics are based on targeting the extracellular domain.”

We appreciate the reviewer for pointing it out. We have now edited the above sentence as follows: “RAGE antagonists are designed to target either the extracellular V domain of the RAGE, which binds to the RAGE substrates (including AGEs and Ab1-42), or the intracellular domain, which modulates the RAGE signaling and downstream events; However, a majority of the potential RAGE antagonist therapeutics are based on targeting the extracellular domain.42

We have also edited similar long sentences in the manuscript for better clarity.

  1. Lane 488 “collage” – should be “collagen”?

We have corrected the typo to “Collagen”.

Round 2

Reviewer 2 Report

I have checked the changes the authors made. I don't have other suggestions to the article. I recommend to accept the article in present form.